# Sequence–Activity Relationship of ATCUN Peptides in the Context of Alzheimer’s Disease

**DOI:** 10.3390/molecules27227903

**Published:** 2022-11-15

**Authors:** Margot Lefèvre, Kyangwi P. Malikidogo, Charlène Esmieu, Christelle Hureau

**Affiliations:** CNRS, LCC (Laboratoire de Chimie de Coordination), 205 Route de Narbonne, BP 44099, CEDEX 4, 31077 Toulouse, France

**Keywords:** ATCUN peptide, copper, reactive oxygen species, kinetics

## Abstract

Amino-terminal Cu^II^ and Ni^II^ (ATCUN) binding sequences are widespread in the biological world. Here, we report on the study of eight ATCUN peptides aimed at targeting copper ions and stopping the associated formation of reactive oxygen species (ROS). This study was actually more focused on Cu(Aβ)-induced ROS production in which the Aβ peptide is the “villain” linked to Alzheimer’s disease. The full characterization of Cu^II^ binding to the ATCUN peptides, the Cu^II^ extraction from Cu^II^(Aβ), and the ability of the peptides to prevent and/or stop ROS formation are described in the relevant biological conditions. We highlighted in this research that all the ATCUN motifs studied formed the same thermodynamic complex but that the addition of a second histidine in position 1 or 2 allowed for an improvement in the Cu^II^ uptake kinetics. This kinetic rate was directly related to the ability of the peptide to stop the Cu^II^(Aβ)-induced production of ROS, with the most efficient motifs being HWHG and HGHW.

## 1. Introduction

Alzheimer’s disease (AD), the most common neurodegenerative disorder, affects more than 30 million people in the world [1] and is characterized by brain deterioration leading to problems with memory, behavior, and thinking [1,2]. According to the amyloid cascade hypothesis [3,4,5], the formation in the synaptic cleft of extracellular senile plaques containing high levels of copper and zinc ions embedded in supramolecular assemblies of the amyloid-β peptides (Aβ) occurs in the AD-affected brain at early stages of the disorder [6,7,8]. Such metal ions can be linked to the etiology of the disease, although they are also essentials and play key biological roles when placed in different biological environments [9]. The depletion of essential copper and zinc pools caused by the sequestration in the senile plaque peptides is a main issue. Another one is due to the redox ability of copper ions that can cycle in biological medium between its cuprous (Cu^I^) and cupric (Cu^II^) states, thus participating in the formation of reactive oxygen species (ROS) from the incomplete reduction of dioxygen to superoxide, hydrogen peroxide, and hydroxyl radical fueled by a physiological reductant such as ascorbate [10,11,12]. We and others have shown that when bound to Aβ, Cu ions retain their ability to catalyze the formation of ROS, which participate in the oxidative stress observed in AD [13,14]. This is one of the reasons why Cu ions are interesting therapeutic targets in AD [15,16,17,18,19,20].

The Aβ peptide has a well-known metal-binding domain located in the first 16 residues (DAEFRHDSGYEVHHQK) of the full-length 40/42-amino-acid-long peptide [21,22]. It includes N-ligands (imidazole histidine (His) Nim, deprotonated amide N^−^, and terminal amine (NH_2_)) as well as O-ligands such as C=O from an amide bond and carboxylate groups (COO^−^). For Cu^II^, two main forms are in equilibrium near neutral pH, with the terminal amine, two Nim groups from the His6 and His13 or His14, and one C=O from an amide bond, more likely that of the Asp1-Ala2 bond, being bound to the Cu^II^ in the predominant form at neutral pH (Figure 1) [23]. The Cu^II^ affinity of Aβ is about 10^9^ M^−1^ at pH 7 [24]. Cu^I^ is linked by two imidazole groups with a preference for the His6/His13 couple [25,26], while the value of the affinity is still under debate and oscillates between 10^7^ and 10^10^ M^−1^ [27,28]. In the case of Zn^II^, the main binding site is composed of two carboxylate functions (mainly from Asp1 and Glu3) and two imidazole groups from His6 and His13 or His14 [29]. The Zn^II^ affinity of Aβ is about 10^5^ M^−1^ at pH 7 [30,31].

Molecules that target Cu in the AD context currently represent an intense field of research. In this context, we and other research groups have focused on various chelation approaches. Highly relevant and complete reviews on this topic can be consulted by the interested reader [15,20,32]. In the present study, we focused on the use of peptide-based ligands that combined several key prerequisites with respect to further therapeutic applications, which include easily tunable Cu^II^ affinity and selectivity as well as the possibility of appending brain delivery sequences [33,34]. The sequences chosen are shown in Figure 2. All eight of the peptides have a free terminal amine and a histidine (His) residue in the third position. This is the prototypical sequence used to create an amino-terminal copper and nickel (ATCUN) binding motif in which the Cu^II^ ion is bound by the N-terminal amine, the proximal nitrogen atom from the histidine (His), and the two deprotonated amide groups in between [35,36,37,38], as also observed in the solid state via X-ray crystallography data [39]. In such an environment, the Cu^II^ center is tightly bound with an affinity of about 10^12^ to 10^14^ M^−1^ depending on the exact sequence and with a low cathodic potential, making it resistant to reduction by the physiologically relevant ascorbate (Asc) [36]. In addition, the ATCUN motif is highly specific to Cu^II^ and Ni^II^, as its name indicates, but not to the softer Zn^II^ ion, which has no ability to induce the deprotonation of the peptide bond. Hence, such peptides are highly specific to Cu^II^ with respect to Zn^II^. Last, they have previously shown to be good candidates to prevent Cu(Aβ)-induced ROS production in the AD context [40,41,42,43,44,45,46]. However, in all but two studies [42,46], the ATCUN peptides were added when the Cu ion was exclusively in its +II state, which is not the biological reality because the brain is quite rich in Asc (about 300 µM) extracellularly [47,48].

We recently showed that when added in presence of a mixture of +I and +II states (i.e., when the Cu(Aβ) is producing ROS), the exact sequence of the ATCUN matters due to participation of Cu^II^ complexation kinetic issues [46]. Within the same context, the importance of such kinetic parameters was previously demonstrated in case of aza-macrocyclic ligands [49]; this originates from the competition between the Cu^II^ removal from Cu^II^(Aβ) by the studied ligand to form a redox stable Cu^II^(ligand) complex and the reduction of Cu^II^(Aβ) to Cu^I^(Aβ) by ascorbate.

In the present work, our main objective was thus to reveal some of the key features that the ATCUN sequences should have to be as appropriate as possible with respect to such kinetic issues. In the series of the eight peptides studied (Figure 2B), several parameters were varied; the main one was the number of His: one or two. Indeed, based on recent results, the addition of a second His was expected to accelerate Cu^II^ binding and the removal from Aβ and to help the coordination of Cu^I^ in a site close to the one observed in Aβ (Figure 1). For one-His-containing sequences, we tested the effect on steric hindrance by varying the two first residues: GGHW (red), GVHW (orange), and VGHW (yellow). For two-His-containing peptides [36,41,46,50,51], we varied the position of the second His (in position 1 or 2) and the position of the Trp residue, resulting in the following four: HGHW (light blue), HWHG (light green), GHHW (navy blue), and WHHG (dark green). In addition, we used the DAHW peptide as a reference because the DAH sequence is found in human serum albumin and has thus been extensively studied [35,36,37,38,52,53]. All of the peptides were amidated at the C-term. A Trp residue was added to monitor the Cu^II^ binding and Cu^II^ removal from Aβ at a concentration in the µM range [54,55], thus closer to the physiological range than the classical concentrations used for spectroscopic monitoring via UV–vis and EPR (in the low mM range). Lastly, note that we used the DAEFRHDSGYEVHHQK sequence (later noted as Aβ for matter of simplicity) isoform, which does not aggregate like the full-length peptides, but as stated previously, retains the metal-binding properties [21,22] and is thus appropriate to study events linked to Cu^II/I^ coordination such as those reported in this work.

First, the spectroscopic and electrochemical descriptions of the Cu^II^ site in the eight ATCUN peptides and the determination of their rates to remove Cu^II^ from Aβ will be reported. Then, we will show that the two-His-containing peptides were the most appropriate to stop Cu(Aβ)-induced ROS production, with HGHW and HWHG regarded as the most promising due to their faster Cu^II^ extraction rate from Aβ. The study of the tryptophan fluorescence quenching using copper was innovatively employed and correlated with the ability of peptides to stop Cu(Aβ)-induced ROS production.

## 2. Results and Discussion

### 2.1. Cu^II^ Binding by the Peptides and Removal from Aβ

The eight ATCUN peptides (referred to as P thereafter) were prepared according to standard solid-phase peptide synthesis (SPPS) procedures on a Rink amide resin following a general synthetic route. Details of peptides synthesis and characterizations are given in the experimental section and Appendix A.

The eight Cu^II^(P) complexes were characterized using UV–vis (Appendix A) and EPR (Figure 1 and Appendix A) spectroscopies; the characteristic parameters are listed in Table 1. The eight complexes had the spectroscopic fingerprints of Cu^II^(ATCUN) complexes; i.e.: (i) a d-d transition near 520 nm with a molar extinction value (ε) of about 100 M^−1^ cm^−1^; and (ii) g-values of about 2.18 (g_//_) and 2.05 (g_ꓕ_) and an A_//_ of about 220 10^−4^ cm^−1^ [35,36,37,38,39]. The recorded EPR spectra were classical for mononuclear square planar complexes with 4N coordination and had g_//_ values higher than g_ꓕ_, characteristic of an elongated Jahn–Teller distortion around the Cu^II^ center. Moreover, superhyperfine lines in the perpendicular region also indicated N equatorial ligands and were reminiscent of Cu^II^ bound in a ATCUN motif [35,36,37,38,39]. These measurements showed that the variations in the ATCUN sequence did not change the first coordination sphere around the Cu^II^ ion but did affect the second sphere, as some minor but relevant changes were observed, especially via EPR (compare, for instance, the spectra of the Cu(GGHW) and Cu(HWHG) complexes, which were less and more sterically burdened, respectively). More precisely, while the g_//_ values remained virtually identical (within the error bar of the measurements), the A_//_ value was increased for the 2-His-containing Cu(P) complexes. This was reminiscent of what has been reported for β-alanine-containing ATCUN [56] or cyclic ATCUN [57] Cu^II^ complexes. It is worth noting that for the GHHW and WHHG peptides, the coordination site composed of three nitrogen atoms (noted 3N) from the N-terminal amine, the imidazole (Im) ring forming the second His, and the deprotonated amide in between was not detected here, in contrast to previous reports on Cu^II^ binding by AHH-COOH peptides [46,51,58]. This may be linked to the amidation of the C-term carboxylate in the present case, which decreased the pH of interconversion between the 3N and 4N (that is to say, ATCUN) coordination modes, in line with the data reported on AHH-CONH_2_ [58].

The Cu^II^(P) complexes were also studied using cyclic voltammetry (CV) (Figure 2, Appendix A). In the cathodic scan, no processes were observed for the Cu^II^ complex made with one-His peptides; one irreversible anodic process was observed at about −1.0 V vs. SCE (−0.76 V vs. NHE) for the four Cu^II^ complexes constructed using the double-His peptides, which was in line with previously reported data [39,42,50,56,59]. For the two-His-containing Cu^II^(GHHW) and Cu^II^(WHHG), an additional cathodic peak was detected at about –0.5 V vs. SCE (0.74 V vs. NHE) (Figure 2 and Appendix A). This peak was at a similar potential value to those detected for the 3N form of the Cu^II^(AHH-COOH) complex [51] or Cu^II^(GHK) [39] and Cu^II^(RHDSG) [60], which adopted a 3N Cu^II^ binding site (Figure 3C). This indicated that although it was not detected by EPR, the 3N form of the Cu^II^(GHHW) and Cu^II^(WHHG) complexes, in which Cu^II^ was coordinated by three equatorial ligands, the NH_2_ of the N-terminus, the N of the imidazole of His, and the nitrogen of the peptide bond between AA1 and AA2 (AA = amino acid residue, NH_2_, N, NIm) as shown in Figure 3C exists in a sufficiently high amount to be reduced and observed by CV. Indeed, as the reduction of the 3N form occurred prior to that of the 4N form in such 3N/4N hybrid complexes, the electrochemical process drives the 3N–4N equilibrium to the 3N form in an electrochemical-chemical process. It was noteworthy that when the second His was in the first position instead of the second position, the CV trace was different, indicating that in the case of a “split His”, as for HGHW and HWHG peptides, the analogues of the 3N form, if existing, were not in a sufficiently high amount to be detected even by CV. For all Cu^II^(P) complexes, such low cathodic potential values indicated that the Cu(P) complexes should resist to reduction by Asc (below a certain concentration).

In the anodic scan, two main processes were expected: the oxidation of the Cu^II^(P) to Cu^III^(P) and that of the Trp residues. None of them could be clearly observed except for Cu^II^(WHHG) and Cu^II^(HWHG), for which an irreversible peak at −0.6 V vs. SCE (−0.84 V vs. NHE) and 0.7 V vs. SCE (0.94 V vs. NHE) were detected, respectively; these were attributed to the oxidation of the Trp residues (Appendix A) [61]. Indeed, we mainly observed the oxidation of the HEPES buffer. Note that we decided to work in HEPES buffer for such studies instead of phosphate buffer even though it has a larger electrochemical window, since it was shown recently that phosphate ions can interact with 3N Cu^II^ species and modify their properties [62]. However, the results indicated that the Cu^II^(P) complexes’ oxidations were at values that were too high for the complexes to be oxidized by O_2_.

After having shown that the peptides were all able to bind Cu^II^ and to form the corresponding ATCUN complexes, we will now describe their ability to remove Cu^II^ from Cu^II^(Aβ). The UV–vis and EPR spectra resulting from the addition of P on Cu^II^(Aβ) are given in Figure 3 and compared to the signatures of the Cu^II^(P) corresponding complexes (for the one-His-containing Cu(GVHW) complex and the two-His containing Cu(GHHW); for the other peptides, see Appendix A). The UV–vis and EPR spectra of Cu^II^(P) and P + Cu^II^(Aβ) were superimposable for all the peptide sequences, showing the ability of all P to remove Cu^II^ from Cu^II^(Aβ). This result was fully in line with the respective Cu^II^ affinity value of ATCUN peptides [35,36,37,38,39] and Aβ [24]. It is nevertheless important to note that for the one-His-containing peptides, the Cu^II^ extraction step from Cu^II^(Aβ) was slow, in contrast to the two-His-containing peptides, which could rapidly extract Cu^II^ from Cu^II^(Aβ); see vide infra.

Finally, the rate of Cu^II^ extraction from Cu^II^(Aβ) by the eight peptides under study was evaluated using Trp fluorescence and its quenching when close to the paramagnetic Cu^II^ ion (Figure 4 and Appendix A). At first glance, the experiment showed that the rate was in the following order: one-His peptides < DAHW—two-His peptides with His in positions 2 and 3 < two-His peptides with His in positions 1 and 3. Actually, a deeper inspection of the data showed that they could not be reproduced using monoexponential curves, which was in line with the presence of several processes at play (vide infra). Several observations could be made: (i) one-His peptides were slower than two His-peptides; (ii) the DAHW peptide had a faster kinetic than the other one-His peptides; (iii) among the two His peptides, those with a “split” His (i.e., HGHW and HWHG vs. GHHW and WHHG) were the fastest ones; and (iv) the position of the Trp in the double-His peptides (position 1 vs. 4 in GHHW and WHHG or position 2 vs. 4 in HWHG and HGHW) had no impact on the kinetics of Cu^II^ extraction from Cu^II^(Aβ) by the peptides.

These observations could be analyzed by considering several factors that influenced the two key steps in the reaction of Cu^II^ removal from Cu^II^(Aβ): the anchoring of the Cu^II^ by the peptides leading to intermediates species and the reshuffling of the intermediates to release the Cu^II^(P) complexes (ATCUN). Indeed, as recently thoroughly described in case of the GGH-COOH peptide [63], the formation of the Cu^II^(ATCUN) complex proceeds via several steps that include the transformation of the 1N intermediates (anchoring via the third His residues) to 2N (binding by the N-terminal amine and the His) (Figure 3A) to 4N (ATCUN motif, Figure 2A) [63,64,65].

In the present case, we deduced that having an amino acid residue with a side chain containing donor atoms in the first position accelerated the anchoring and the formation of the intermediates. In the case of the DAHW, Cu^II^ anchoring can be proposed to occur through a 1N1O brace (Figure 3D) and for HGHW and HWHG via a 2N brace (Figure 3E). We hypothesized that the 2N brace is a better motif for the Cu^II^ coordination that would explain the different kinetics of Cu^II^ capture.

Among the other peptides, the GHHW and WHHG peptides were the fastest, as expected based on the literature, due to the formation of a 3N intermediate composed of the N-terminal amine, the His side-chain in position 2, and the deprotonated peptide bond in between, as recently described for AHH-COOH (Figure 3B) [36,46].

### 2.2. Arrest of Cu and Cu(Aβ)-Induced ROS Production by the Peptides

We will now study the effects of the eight peptides on Cu and Cu(Aβ)-induced ROS production. The Asc reductant fueled the reaction of incomplete dioxygen reduction to O_2_°^−^, H_2_O_2_, and HO° [10]. Hence, monitoring its consumption rate was an easy and robust way to monitor the formation of ROS. The concentration of Asc that fueled the reaction was followed by UV–vis at 265 nm (λ_max,_ ε = 14 500 M^−1^ cm^−1^). We previously showed that Asc consumption mirrors H_2_O_2_ formation (and to a lesser extent, HO° formation) [66,67]. Note that with respect to HO°, the correlation with Asc consumption is only qualitative because HO° is able to react with the Aβ peptide itself and thus it is not HO° formation but actually HO° release from the Cu(peptide) complex that is monitored [68,69]. Asc consumption monitoring is thus now regarded as a classical method to evaluate ROS production by Cu^I/II^ complexes [41,70,71,72].

Three experiments were performed: (1) the addition of P to a Cu^II^ (or Cu^II^(Aβ))-only species and then the addition of Asc to trigger the formation of ROS; (2) the addition of P after the addition of ascorbate, leading to the presence of both Cu^I^ and Cu^II^ species (or Cu^I^(Aβ) and Cu^II^(Aβ)) in the medium; and (3) similar to (2) but starting with Zn^II^(Aβ) instead of Aβ. In case (1), the experiment documented the ability of P to bind Cu^II^ (or remove it from Cu^II^(Aβ)) and to form a Cu^II^(P) complex resistant to reduction by Asc. In case (2), not only the thermodynamic ability of P to bind Cu^II^ (or remove it from Cu^II^(Aβ)) was probed but also the velocity at which the reaction was performed. Indeed, if the Cu^II^ binding was too slow, the reduction would proceed and the ROS would not be fully stopped. The extent of the lessening of the Asc consumption depended on the relative rates between the Cu^II^ chelation and Cu^II^ reduction reactions. In case (3) the experiments probed several key features; that is to say, the ability to remove Cu^II^ from Aβ in presence of Zn^II^ (Cu^II^ over Zn^II^ selectivity of P versus Aβ) and to form the Cu^II^(P) (Zn^II^ impact on the kinetics of Cu^II^ removal from Cu^II^(Aβ)).

The experiments at the Cu^II^ level (case (1)) with or without Aβ being present are shown in Figure 5A,B and Appendix A, respectively. In the absence of Aβ with all the peptides and with an incubation of 30 s before the addition of Asc, its consumption was fully prevented (Appendix A). In presence of Aβ with all the peptides and with an incubation of 30 min before the addition of Asc, its consumption was fully prevented (Figure 5B). This indicated that all the peptides were able to remove Cu^II^ from Cu^II^(Aβ) and that the formed Cu^II^(P) species could not be reduced by Asc in such conditions, which was in line with other experiments and the electrochemical data, respectively. It is worth noting that when the incubation time was reduced to 30 s, only the two-His-containing (and to a lesser extent, the DAHW) peptides retained their ability to prevent Asc consumption (Figure 5A). This indicated that the complexation reaction was not fully completed by the other three peptides.

The experiments in which both Cu^I^/Cu^II^ were present (case (2)) with or without Aβ being present are shown in Figure 5C and Appendix A, respectively. The peptides behaved differently with two main trends observed: either the instantaneous arrest of Asc consumption for the four two-His-containing peptides and the DAHW; or an initial slowdown to eventually reach the basal rate of Asc consumption, with GGHW being more rapid than VGHW, which itself was more rapid than GVHW. Similar observations included the signatures of a kinetically controlled binding of Cu^II^, which was previously observed with a nonpeptide ligand in the presence of Aβ [49,67]. In the presence of Aβ, the differences between the peptides were more pronounced, with DAHW becoming less efficient (only a slowdown in Asc consumption was observed, not a full arrest) and the other three one-His peptides being unable to arrest Cu^II^(Aβ)-induced Asc consumption. This higher discrimination between the peptides could have two origins: (i) the time requested to extract Cu^II^ from Aβ was higher than to bind Cu^II^ due to pre-equilibrium between Aβ-bound and Aβ-unbound Cu^II^ and/or due to a more difficult anchoring for Aβ-bound Cu^II^ via the formation of a ternary species; and/or (ii) the ratio between Cu^I^ and Cu^II^ was shifted toward Cu^I^ in presence of Aβ, leading to a weaker level of the targeted Cu^II^. Since the relative efficiency of the peptides perfectly matched the rate of Cu^II^ removal from Aβ as measured previously by Trp-fluorescence quenching experiments, we anticipated that the first effect (i) was predominant.

In the presence of Zn^II^ (Figure 5D), the effects of the peptides were not prevented, which was in line with the relative Cu^II^-over-Zn^II^ selectivity of the peptides and Aβ, and were even slightly improved for the one-His peptides. This may have indicated that Zn^II^ helped the Cu^II^ dissociate from the Cu(Aβ), as previously described for the parent Cu(AAH, AHH) complexes [46], thus increasing the level of accessible Cu^II^.

A final type of experiment was performed in which the Asc consumption was triggered by the addition of air to a premixed solution containing Cu^I^ (Appendix A) or Cu^I^(Aβ) (Appendix A) (coming from the reduction of Cu^II^ and Cu^II^(Aβ) by Asc under argon) and the peptides. Similar results to those obtained in the presence of both Cu^I^ and Cu^II^ redox states were obtained regardless of the presence of Aβ, with more pronounced differences between the one-His-containing peptides. This was in line with the higher level of Cu^I^ in this latter experiment compared to the previous one (Cu^I^/Cu^II^ and addition of peptides during the course of Asc consumption—Appendix A).

## 3. Concluding Points

A first main point to be underlined is the fact that in contrast to the current paradigm, not all ATCUN-forming peptides could stop Cu-induced ROS production. When the Cu ions were engaged in the redox cycle (i.e., when Cu^I^ was present in addition to Cu^II^), some ATCUN sequences failed to stop Cu-induced ROS production; this trend worsened in the presence of Aβ.

The efficiency of the peptides to arrest the Cu^II^(Aβ)-induced ROS production perfectly followed the rate of Cu^II^ extraction from Cu^II^(Aβ). This was quite remarkable and indicated that for this series of peptides, the rate of Cu^II^ extraction from Cu^II^(Aβ) was the determinant step in the arrest of Cu(Aβ)-induced ROS production.

In Figure 4, the two possible paths to arrest Cu^II^(Aβ)-induced ROS are shown: when Cu^I^(Aβ) and Cu^II^(Aβ) were cycling, extraction of Cu^II^ from Cu^II^(Aβ) (path Cu^II^, in pink) or from Cu^I^(Aβ) led to the formation of a Cu^I^(peptide) complex that was oxidized to Cu^II^(peptide) complexes, which could not be reduced back (path Cu^I^ in blue). Our data indicated that the Cu^II^ path was the one in play for the peptides studied here. This was not fully anticipated and a Cu^I^ path was first foreseen, especially in the case of the two-His-containing peptides that could compete with Aβ to bind Cu^I^, having *a priori* the same Cu^I^ site (Figure 1) [42,73].

The HGHW and HWHG peptides were faster in capturing Cu^II^ out of Cu^II^(Aβ) and worked perfectly in all of the experimental conditions tested in the Asc consumption assay (from Cu^I^ in the presence of Zn^I^). The possibility of having intermediate complexes with Cu^II^ bound by the N-terminal amine and its imidazole ring side chain (thus forming a metallacycle) and by the side chain of the His in position 3 (3N-brace site, Figure 3D), which was reminiscent of the histidine brace LPMO binding site in Ref. [74], is proposed to be the reason why. It is also worth noting that the bulkier Trp residues in position 2 had no impact on the ability of the peptide to arrest Cu^II^(Aβ)-induced ROS production. The quenching of the Trp fluorescence was thus a fully appropriate tool to evaluate the ATCUN motif’s ability to stop Cu^II^(Aβ)-induced ROS production. Nevertheless, for the two-His-containing peptides this technique did not allow us to probe the early steps of the Cu^II^ coordination, which were too fast to be measured. More advanced stopped-flow and freeze-quenched techniques could be used, as recently reported [63,64].

In line with previous reports [32,40,41,46], ATCUN sequences were revealed to be good candidates for further therapeutic purposes in the Cu^II^ chelation field. However, when Cu^II^-induced or Cu^II^(Aβ)-induced ROS are targeted, the exact sequence matters.

## 4. Experimental Section

All of the chemicals were purchased from Sigma-Aldrich unless otherwise specified. All of the solutions were prepared using ultrapure water (18.2 MΩ). Stock solutions of metallic salts and peptides were prepared by dissolving the salts or the peptides in ultrapure water; the concentrations were determined using UV–vis absorption spectroscopy. A stock solution of Zn^II^ ions was prepared at 100 mM using monohydrated ZnSO_4_·H_2_O salt. A stock solution of Cu^II^ was prepared at 100 mM using hydrated CuSO_4_.5H_2_O salt. The concentration of the solution was determined using UV–vis (λ = 800 nm corrected at 400 nm) while considering a molar extinction coefficient of 12 M^−1^ cm^−1^. A stock solution of HEPES buffer (sodium salt of 2-[4-(2-hydroxyethyl)piperazin-1-yl]ethanesulfonic acid) was prepared at 500 mM and the pH was adjusted to 7.4 using concentrated sodium hydroxide. All pH values are given with a ±0.1 pH unit error. A stock solution of sodium ascorbate was prepared at 5 mM every two days due to its degradation in solution. The Aβ_1-16_ (DAEFRHDSGYEVHHQK) peptide was purchased from Genecust (purity > 95%). A stock solution was prepared at around 10 mM; its concentration was determined using UV–vis with Tyr10 absorption considered as free tyrosine (in acidic condition, ε276−ε360 = 1410 M^−1^ cm^−1^). The ATCUN peptides were synthetized in house and stock solutions were prepared at around 10 mM. The peptide concentration was determined by a Cu^II^ titration followed by UV–vis or tryptophane residue absorption (ε280 = 5600 M^−1^ cm^−1^). The stock solutions were stored at 4 °C and the Aβ peptide and ATCUN peptide stock solutions at −20 °C.

**Synthesis of peptides**. The ATCUN peptides were synthesized using a Liberty Blue microwave peptide synthesizer with a standard 9-fluorenylmethoxycarbonyl (Fmoc) strategy on a 4-(2′,4′-dimethoxyphenyl-Fmoc-aminomethyl)-phenoxyacetamido-norleucyl-MBHA resin (Fmoc-Rink Amide MBHA resin from Fluorochem, 0.33 mmol/g loading, 1% DVB, 100–200 mesh) through solid-phase peptide synthesis protocols [75]. The coupling reactions were performed by using a 5-fold excess of amino acid, 5-fold excess of DIC, and 5-fold excess of Oxyma in DMF. The N-terminal Fmoc deprotection was carried out using 20% piperidine in DMF. Resin cleavage and side-chain deprotection were performed at the same time via treatment with 95% TFA, 2.5% H_2_O, and 2.5% triisopropylsilane (TIS) for 180 min. The peptides were precipitated with cold ether from the solution after cleavage, centrifuged, dissolved in H_2_O containing 0.1% TFA solution, and purified via ChromatoFlash on a INTERCHIM PURIFLASH BIO C18-T (pore size 200 Å, particle size 15 μm, 25 g) column using a gradient from 95% to 50% (*v*/*v*) of H_2_O with 0.1% TFA and MeOH with 0.1% TFA. Pure peptides were lyophilized. The purity of the peptides was assessed via NMR and (+)ESI-MS.

**NMR spectroscopy**. The ^1^H and ^13^C experiments were recorded on a Bruker Avance III 400 MHz spectrometer equipped with a 5 mm broadband inverse triple-resonance probe 1H, BB (31P-103Rh)/31P with Z field gradients. All spectra were calibrated with respect to the D_2_O signal (4.79 ppm). The chemical shifts (δ) are reported in ppm. The NMR coupling constants (J) are reported in Hz.

**UV–visible spectroscopy**. Titration and UV–vis kinetic data from the ascorbate consumption experiments were recorded with a Hewlett Packard Agilent 8453 or 8454 spectrophotometer at a controlled temperature of 25 °C in a 1 cm path length quartz cuvette with 800 rpm stirring.
-**Cu^II^ titration**. The peptide (P) solutions’ precise concentrations were determined via Cu^II^ titration with a solution of known concentration using the d−d transition absorption of the complex to determine the equivalence point or following the *λ_max_* of tryptophane at 280 nm using the molar extinction coefficient of the tryptophan (ε = 5600 M^−1^ cm^−1^). P were titrated at 500 µM with the addition of an increasing amount of a Cu^II^ stock solution (by 0.1 eq). The UV–vis titration experiments confirmed the stoichiometry of the metal–ATCUN peptide complex at 1:1.-**ROS experiment.** The ROS production was determined using an ascorbate consumption assay monitored via UV–vis spectroscopy. The decrease in the absorption band at *λ*_max_= 265 nm of the Asc (ε = 14,500 M^−1^ cm^−1^, corrected at 800 nm) was plotted as a function of time. The samples were prepared from stock solutions at 1 mM and mixed in situ in the UV–vis cuvette at a final concentration of 12 µM for the ATCUN peptide, Aβ_1-16_, and Zn^II^ and 10 µM for Cu^II^ in HEPES at pH 7.4. Ascorbate was added to obtain 100 µM as the final concentration. The final volume was adjusted with ultrapure water to 2 mL. The ROS experiments were performed following three different procedures: starting from Cu^II^, starting from a mixture of Cu^I^ and Cu^II^, and starting from Cu^I^.

For the Cu^II^ experiment, the ATCUN peptide (P) was added to the Cu^II^ or Cu^II^ + Aβ mixture, and then the ascorbate was introduced in the cuvette. Cu^II^, Aβ, P, and ascorbate were added at 30, 60, 120, and 180 s, respectively. For some experiments, the Cu^II^ + Aβ + P mixture was preincubated for 30 min and added directly into the cuvette containing the aqueous buffered solution. The ascorbate was then introduced at 180 s.

For the Cu^I^ and Cu^II^ experiment, the ascorbate was introduced first into the cuvette; then, either Cu^II^ + P or Cu^II^ + Aβ + P or Cu^II^ + Aβ + Zn^II^ + P was added. Ascorbate, Aβ, and/or Zn^II^ and Cu^II^ were added at 30, 120, and 240 s, respectively. When the absorbance reached about 1.1 in O.D., the ATCUN peptide was added. These experiments were run under aerobic conditions.

Finally, for the Cu^I^ experiment, P was injected with a Hamilton syringe into the Cu^I^ or Cu^I^ + Aβ mixture in a sealed UV–vis cuvette under anaerobic conditions and then exposed to air (by bubbling air into the cuvette). The Cu^I^ was generated from the in situ reduction of Cu^II^ with ascorbate. All of the solutions were previously degassed for 15 min with argon before being introduced into the sealed UV−vis cuvette kept under argon.

**EPR.** Electron paramagnetic resonance (EPR) spectra were recorded using an Elexsys E-500 Bruker spectrometer operating at a microwave frequency of approximatively 9.5 GHz. The spectra were recorded using a microwave power of 5 mW, a magnetic field range of 2400 to 3700 G, and a modulation amplitude of 5 G. The experiments were carried out at 120 K using a liquid nitrogen cryostat. EPR samples were prepared in Eppendorf tubes from a 10 mM Cu^II^ stock solution and diluted to 500 µM in 50 mM of HEPES buffer (pH 7.4) with the addition of 500 µM ATCUN peptides. As a cryoprotectant, 10% glycerol was added. The final volume was adjusted to 200 µL using ultrapure water. The mixture was then transferred into a EPR quartz tube and frozen in liquid nitrogen.

**Fluorescence spectroscopy.** The kinetics of Cu^II^ chelation by the ATCUN peptides were monitored by fluorescence following the tryptophan fluorescence emissions using a Horiba Fluoromax 4 fluorescence spectrophotometer in a 2 × 10 mm path quartz cuvette with *λ*_ex_ = 280 nm and *λ*_em_ = 350 nm. The fluorescence intensity was measured using the following parameters: excitation slit: 2 nm, emission slit: 10 nm, averaging time: 0.1 s. Cu^II^ -binding kinetics that induced the tryptophan fluorescence quenching were measured via the addition of 1 eq of Cu^II^(Aβ) to a solution of the ATCUN peptides (1 µM) in HEPES buffer (100 mM, pH 7.4) and the fluorescence monitored over time (600 s). The normalization of the fluorescence intensity was realized according to the following equation y = F/(Fmax-20,000) for comparison. We decided to take 20,000 and not the F0 because all the curves had a different ending point and the minimum fluorescence intensity reached was 20,000 u.a. The t_1/2_ was calculated from 50 to 70 s during the slow phase of the kinetics.

**Cyclic voltammetry.** The cyclic voltammetry (CV) experiments were performed on an Autolab PGSTAT302N potentiostat controlled with General Purpose Electrochemical System (GPES) version 4.9, Eco Chemie B. V. Ultrecht, The Netherlands. A three-electrode setup was used that consisted of a glassy carbon disk (3 mm diameter) as the working electrode, a saturated calomel electrode as the reference electrode, and a platinum wire as the counter electrode in an argon-flushed 2 mL cell. The working electrode was polished before each measurement on a red disk NAP (Struers) with 1 µM AP-A suspension and then a 0.3 µM AP-A suspension. The sample solutions were degassed for 3 min for each measurement. The scan rate was 0.1 V s^−1^. Three scans were realized for each experiment; only the first scan is shown in the figures.

## Data Availability

Data are available on request from the corresponding author.

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
