# Peer review of "Sequence–Activity Relationship of ATCUN Peptides in the Context of Alzheimer’s Disease"

_molecules, 2022, doi:10.3390/molecules27227903_

Round 1
Reviewer 1 Report
The manuscript by Hureau and Co-workers describes a systematic study of eight ATCUN peptides aimed at targeting copper ions and stop associated formation of Reactive Active Oxygen Species (ROS). In addition, a correlation between the peptide sequences and their impact on Cu(Ab) induced ROS production is drawn. These data are also systematically compared to available results which give credit to the conclusion of the manuscript. In my opinion, the paper is original, interesting, and well-written; thus, it should be suitable for publication after attention to the minor points below.
1) The author reported the EPR data but does not mention the probable geometry adopted from the epr data.
2) The value of CV should be expressed in terms of NHE instead of SCE.
3) Page no 147, The author reported that “For the two-His containing CuII(GHHW) and CuII(WHHG) an additional cathodic peak is detected at about -0.5 V vs. SCE”.
Which redox couple of copper indicates this peak?
4) Fig 7, the Y-axis was written as Abs265-800 nm. it should be only Abs265 nm.
5) The copper transformation occurs in several intermediated pathways by designed peptides. But what effect on the Cu(Ab) geometry during this process?
Author Response
REVIEWER 1.
Comments and Suggestions for Authors
The manuscript by Hureau and Co-workers describes a systematic study of eight ATCUN peptides aimed at targeting copper ions and stop associated formation of Reactive Active Oxygen Species (ROS). In addition, a correlation between the peptide sequences and their impact on Cu(Ab) induced ROS production is drawn. These data are also systematically compared to available results which give credit to the conclusion of the manuscript. In my opinion, the paper is original, interesting, and well-written; thus, it should be suitable for publication after attention to the minor points below.
Author reply:
We thank the reviewer for these overall very positive comments on our manuscript and will answer point by point to the minor points asked.
- The author reported the EPR data but does not mention the probable geometry adopted from the epr data.
Author reply:
We thank the reviewer for this comment and had accordingly a sentence in the text that specify the geometry of the Cu complexes. “The recorded EPR spectra are classical for mononuclear square-planar-complexes with 4N coordination. Moreover, superhyperfine lines in the perpendicular region indicate also N equatorial ligands and are reminiscent of CuII bound in a ATCUN motif”
- The value of CV should be expressed in terms of NHE instead of SCE.
Author reply:
The reduction and oxidation potentials of the complexes have been added vs NHE into the text.
3) Page no 147, The author reported that “For the two-His containing CuII(GHHW) and CuII(WHHG) an additional cathodic peak is detected at about -0.5 V vs. SCE”.
Which redox couple of copper indicates this peak?
Author reply:
The cathodic peak at -0.5 vs SCE is attributed to the reduction of the 3N form of the complex, meaning that the CuII is coordinated by three equatorial ligands, the NH2 of the N-terminus, the N of the imidazole of His and the nitrogen of the peptide bond between AA1 and AA2 (NH2, N, NIm) as shown in scheme 3C.
The text has been modified for more clarity
4) Fig 7, the Y-axis was written as Abs265-800 nm. it should be only Abs265 nm.
Author reply:
We changed the figure 7 (now Figure 5) replacing Abs265-800 nm by Abs265 nm. The caption has been modified, we have added “followed by UV-visible spectroscopy at 265 nm with a background correction at 800 nm”
5) The copper transformation occurs in several intermediated pathways by designed peptides. But what effect on the Cu(Ab) geometry during this process?
The extraction of Cu from CuII(Aβ) can proceed via two mechanisms one been associative and the other one been dissociative. In the associative mechanism a ternary species, CuII(Aβ)-P, is formed. We don’t have any experimental data supporting the formation of such a ternary species. In the dissociative pass, the Cu is first released from CuII(Aβ) and then chelated by P. Thus, the CuII(Aβ) geometry should not have an important impact on the intermediate CuIIP formed.
Reviewer 2 Report
Dear authors,
The manuscript entitled "Sequence-Activity Relationship of ATCUN-Peptides in the Alzheimers’ Disease Context” report on a systematic study of eight ATCUN peptides aimed at targeting copper ions and stop associated formation of Reactive Active Oxygen Species (ROS). It presents scientific relevance for the area of Medicine, Biology and Chemistry area. After consulting www.sciencedirect.com and https://pubmed.ncbi.nlm.nih.gov/, two authors have publications related to subjects related to the theme of the manuscript. The language (English) are satisfactory (I suggest the final revision)! However, you need to change some details/information in the abstract, Introduction, Methods, results, discussion and conclusions. I request information on the procedures and interpretation of the results obtained.
1. Abstract: Adequate, but I suggest rewrite and add information:
- The abstract is well written, with details of the methods used. However, I suggest reducing the information on "methods" and inserting the results obtained, more relevant.
- In line 8, what do the authors indicate as a "systematic study"? This terminology is mostly used for "systematic" reviews!
- I suggest highlighting the "innovative" proposal of the study, as well as the advantages / disadvantages, at the end of the abstract.
- The keywords “copper”, “reactive oxygen species” and “kinetics” are not included in the title or abstract. I suggest review!
2. Introduction section: It is well written, but I suggest:
- Page 2, lines 50-77 and 78-97: Long paragraph! I suggest splitting!
- I suggest presenting figure 1 immediately after its citation in the text! Idem for figure 2 and all figures!
- I suggest highlighting the objective and "innovative" proposal of the study, as well as the advantages / disadvantages, at the end of the introduction.
3. Results section (or “Results and discussion”)?
Wouldn't it be more interesting to combine the "results” with the "discussion" to better describe the findings and compare them with other works published in the literature?? I suggest expanding the discussions!
- I suggest indicating all figures closer to their citation in the text.
- I did not find figures S1 to S7 in the text!!!
- I suggest that "schemes" be indicated as "figures" or supplementary material.
- Page 6, lines 183-184: The aurhors wrote: “...(dependent on the peptide sequences and concentrations at which the experiments are performed)”. I suggest further discussing this information. I suggest expanding the discussions on the data obtained in Figure 5.
- Figure 6 must be inserted after its citation in the text!
3. Concluding points section
- I suggest that this section should be cited after the "Experimental Section", as these are final considerations about the study.
- I suggest rewriting, improving the conclusions based on some comments. I suggest highlighting the advantages of the method and the study! At the end of the section, I suggest highlighting the importance of the study area and limitations!
4. Experimental Section: The methodological proposal is appropriate to the manuscript, but I suggest:
- Page 13, line 376, in “Synthesis of Peptides” section: Did the authors follow any references for this step? If yes, indicate the reference! I suggest reviewing this at all steps/subsection!
- How were the "optimal conditions" determined/optimized for the Fluorescence Spectroscopy and Cyclic Voltammetry steps? How was the validation of the proposed methods carried out?
* Tables and Figures: Adequate! Please see comments above.
* References: Please, check if the references are in accordance with the journal's rules.
Author Response
REVIEWER 2.
The manuscript entitled "Sequence-Activity Relationship of ATCUN-Peptides in the Alzheimers’ Disease Context” report on a systematic study of eight ATCUN peptides aimed at targeting copper ions and stop associated formation of Reactive Active Oxygen Species (ROS). It presents scientific relevance for the area of Medicine, Biology and Chemistry area. After consulting www.sciencedirect.com and https://pubmed.ncbi.nlm.nih.gov/, two authors have publications related to subjects related to the theme of the manuscript. The language (English) are satisfactory (I suggest the final revision)! However, you need to change some details/information in the abstract, Introduction, Methods, results, discussion and conclusions. I request information on the procedures and interpretation of the results obtained.
Author reply:
We thank the referee for His/her comments. We answered point by point to the comments below. We did consider his/her suggestions to clarify the novelty and biological outcomes of our study. We do also think that our paper is perfectly suited for a chemistry journal such as Molecules. Thus, we try to reach the best compromise between the referee’s requests to push the originality too much and the proper positioning of our work.
- Abstract: Adequate, but I suggest rewrite and add information:
- The abstract is well written, with details of the methods used. However, I suggest reducing the information on "methods" and inserting the results obtained, more relevant.
Author reply:
Changes have been made based on the referee comment’s. The abstract is now more relevant including the results obtained. We do think that the abstract now meet the criterium “In other words, give a background and motivation to the paper, a brief description of the methods, the principal results, and then conclusions or interpretations” as edited by MDPI
- In line 8, what do the authors indicate as a "systematic study"? This terminology is mostly used for "systematic" reviews!
Author reply:
The term “systematic” has been removed for clarity.
- I suggest highlighting the "innovative" proposal of the study, as well as the advantages / disadvantages, at the end of the abstract.
Author reply:
The abstract has been rewritten and we do think that it meets now the criterium edited by MDPI : “In other words, give a background and motivation to the paper, a brief description of the methods, the principal results, and then conclusions or interpretations”
- The keywords “copper”, “reactive oxygen species” and “kinetics” are not included in the title or abstract. I suggest review!
Author reply:
Key words are now included in the abstract.
- Introduction section: It is well written, but I suggest:
- Page 2, lines 50-77 and 78-97: Long paragraph! I suggest splitting!
Author reply:
The sentence has been divided as asked by the reviewer.
- I suggest presenting figure 1 immediately after its citation in the text! Idem for figure 2 and all figures!
Author reply:
We did our best to present the figures (and schemes) at more appropriate places in the revised version of our manuscript.
- I suggest highlighting the objective and "innovative" proposal of the study, as well as the advantages / disadvantages, at the end of the introduction.
Author reply:
A sentence has been added at the end of the introduction as suggested by the referee. “For the first time, the study of the tryptophan fluorescence quenching by copper in an ATCUN containing sequence was innovatively employed and correlated with the ability of peptides to stop Cu(Aβ)-induced ROS production.”
- Results section (or “Results and discussion”)?
Wouldn't it be more interesting to combine the "results” with the "discussion" to better describe the findings and compare them with other works published in the literature?? I suggest expanding the discussions!
Author reply:
We thank the reviewer for this observation and have changed accordingly the section “results” into “results and discussion” which is more representative.
- I suggest indicating all figures closer to their citation in the text.
Author reply:
Changes have been made accordingly.
- I did not find figures S1 to S7 in the text!!!
Author reply:
We do refer to those figures in the main text, some examples are listed here after.
S1: page 4 line l22, page 6 line 181,
S2 page 6 line 181
S3 and S4: page 5 line 144
S5 page 8 line 207
S6 page 11 277
S7: page 11 line 313
- I suggest that "schemes" be indicated as "figures" or supplementary material.
Author reply:
The term Scheme is used here on purpose, in the revised version we changed Figure 1 and 2 to scheme 1 and 2 respectively for uniformity.
- Page 6, lines 183-184: The aurhors wrote: “...(dependent on the peptide sequences and concentrations at which the experiments are performed)”. I suggest further discussing this information. I suggest expanding the discussions on the data obtained in Figure 5.
Author reply:
The discussion has been rewritten to meet the suggestion of the referee. “The UV-vis and EPR spectra of CuII(P) and P + CuII(Aβ) are superimposable for all the peptide sequences showing the ability of all P to remove CuII from CuII(Aβ). This result is fully in line with the respective CuII affinity value of ATCUN peptides35-39 and Aβ.24 It is nevertheless important to note that for the 1His containing peptides, the CuII extraction step from CuII(Aβ) is slow, in contrast to the 2His containing peptides which can extract rapidly CuII from CuII(Aβ), see vide infra.”
- Figure 6 must be inserted after its citation in the text!
Author reply:
It has been done
- Concluding points section
- I suggest that this section should be cited after the "Experimental Section", as these are final considerations about the study.
Author reply:
Most of the papers published in Molecules present first the conclusion and after the “Experimental Section” and for a better flow of the reading, we have decided to keep the order “Concluding points section” and then “experimental section”.
- I suggest rewriting, improving the conclusions based on some comments. I suggest highlighting the advantages of the method and the study! At the end of the section, I suggest highlighting the importance of the study area and limitations!
Author reply:
The method used in our study are classical methods used in bio-inorganic chemistry in the field of copper chelator in AD context and do not need in our opinion to be highlighted.
Moreover, the importance of the study is clearly stated in the conclusion see for example the sentences: (1) “A first main point to be underlined is the fact that in contrast to the current paradigm, not all ATCUN forming peptides can stop Cu-induced ROS production” and (2) “ATCUN sequences reveal to be good candidates for further therapeutic purposes in the CuII chelation field. But when CuII-induced or CuII(Aβ)-induced ROS are targeted, the exact sequence matters !”
The limitation of the study have been highlighted “. The quenching of the Trp fluorescence is thus a fully appropriate tool to evaluate the ATCUN motif ability to stop the CuII(Aβ)-induced ROS production. Nevertheless, for the 2His containing peptides it doesn’t allow to probe the early step of the CuII coordination which are too fast to be measured. Some more advanced stopped-flow and freeze quenched techniques could be used as recently reported.”
- Experimental Section: The methodological proposal is appropriate to the manuscript, but I suggest:
- Page 13, line 376, in “Synthesis of Peptides” section: Did the authors follow any references for this step? If yes, indicate the reference! I suggest reviewing this at all steps/subsection!
Author reply:
A reference has been added for the synthesis of peptide
- How were the "optimal conditions" determined/optimized for the Fluorescence Spectroscopy and Cyclic Voltammetry steps? How was the validation of the proposed methods carried out?
Author reply:
For the fluorescence spectroscopy the parameters were chosen after some trials: Conditions were optimized in term of concentration. Indeed, to be able to see the quenching of fluorescence the process must be slow enough. To do that it is possible to adjust the concentration. Less the sample is concentrated, slower is the reaction. For this reason, 1 µM was chosen. Different sizes of slit have been used, to achieve a high fluorescence intensity but below the saturation threshold of the detector to follow the quenching as well as possible.
Cyclic voltammetry experiments were very classically conducted for the bio-inorganic chemistry field and more precisely for copper complexes, see for example ref Chem.Eur.J.2021,27,1777 –1786. Moreover, as mentioned in the text HEPES buffer was chosen instead of phosphate buffer for the following reason “Note that we decided to work in HEPES buffer for such studies instead of phosphate buffer although it has a larger electrochemical window, since it was shown recently that phosphate ions can interact with 3N CuII species and modify their properties (Dalton Transactions 2021, 50, 2726-2730)”
We let the referee or the editorial staff decide whether it is appropriate to include these commentaries into the manuscript.
How was the validation of the proposed methods carried out?
Author reply:
The proposed methods were validated according to literature and PI appreciation and knowledge. The methods chosen were those giving robust results and as close as possible to data from literature for an easier comparison.
Tables and Figures: Adequate! Please see comments above.
* References: Please, check if the references are in accordance with the journal's rules.
Author reply:
The references are now in accordance with the journal’s rules.
Reviewer 3 Report
Please find the attached report.

Author Response
REVIEWER 3.
The manuscript by Hureau et. al. describes sequence-activity relationship of ATCUN-peptides in the Alzheimers’ disease context. They have demonstrated it through 8 ATCUN peptide sequences and their binding to CuII and ability of CuII extraction from CuII(Aβ) peptides to prevent ROS formation catalyzed by CuII(Aβ). Although the novelty of the work is not significant, authors have tried to assign general requirements for rationalize ATCUN peptide fragments required to form a strong complex selectively with Cu and Ni. They have emphasized this as a strategy to stop Cu-induced ROS production, especially in AD. ATCUN sequences reveal to be good candidates for further therapeutic purposes in the CuII chelation field.
The Manuscript is complex yet very well written. Overall, this manuscript may be considered for the publication in this journal after below mentioned corrections and suggestions to improve the paper.
We thank the referee for his/her comments, point by point answer to the referee is given below.
Writing: Overall size of the manuscript can be made more concise.
Author reply:
We are sorry but the manuscript cannot be more concise as we have to expend some sections as asked by reviewer 2.
Page 1, line 41: Add space and fullstop. ------and carboxylate groups 40 (COO-). For-----
Author reply:
The change has been made
Page 2, line 64: what is Asc? Please provide its long form when it appeared for the first time.
Author reply:
The long term is now in the text
Page 3, line 101: ----to stop Cu(Aβ)-induced ROS production while HGHW and HGHW. Authors has repeated the same sequence twice.
Author reply:
We thank the reviewer for this observation and we have replaced one of the HGHW by HWHG.
Page 3, line 107: what is Ri?
Author reply:
The caption of the figure 2 (now scheme 2) has been modified.
Page 3, Figure 2: Please number the positions (I, 3 etc) in panel A.
Author reply:
The change has been made
Page 3, Figure 2: Please use consistent numbering to indicate the position (for example I, II, III OR 1, 2, 3).
Author reply:
A consistent numbering is now used
Page 4, Figure 3: Please indicate (highlight) the main difference between 2H peptides Vs 1H peptide. Is is the minor reduction of g-factor at 2.1-2.2?
Author reply:
As mentioned in the manuscript the first coordination sphere of the complexes is the same for the 2H peptides Vs 1H peptide. The slight differences of the spectra observed are due to a difference in steric hindrance. It is known that an enhanced tetrahedral distortion within the CuN4 system will give rise to a value of gz increasing and Az decreasing. The 2H peptides have a higher A value that would suggest that the geometry of the complex is more square planar around the Cu and less distorted that with 1H. This has been emphazised in the text accordingly: “More precisely, while the g// values remains virtually identical (within the error bar of the measurements), the A// value is increased for the 2-His containing Cu(P) complexes.”
- Page 4, Figure 3, line 119: what are rationales to use specifically CuII(GGHW) complex as a reference? Because earlier it was mentioned that the peptide DAHW is the refence peptide.
Author reply:
DAHW is indeed the reference as mentioned is the text because it is the ATCUN motif which is found biologically. From a chemical point of view GGHW is simplest motif, it is the reason why it has been chosen as reference. This has been modified in the caption of Figure 1. (“complex arbitrarily chosen as the internal reference”).
Page 4, Figure 3, 120: concentration of [P] 500 uM however in the experimental details, concentration is 600 uM. Please mention the right concentration.
Author reply:
The modification has been made
Page 4, Figure 3: Measurement of control is missing. it is recommended to add g-factor curve for (i) refence peptide CuII(DAHW) and one without Cu in it
Author reply:
In page 4, the figure 3 (now Figure 1) contains already the CuII(DAHW) curve asked by the reviewer. The control experiment (without CuII) has been performed to check any possible contamination by paramagnetic species and as expected show no signal.
.
Page 4, Figure 3: Is there any scale on y-axis to indicate the intensity of the peaks?
Author reply:
The y-axis is the second derivatives of the absorption, this information has been added to the figure caption.
Page 4, line 126: what is the difference between gI and gII.
Author reply:
The g-factor for anisotropic complexes is usually split into three values of g along a cartesian coordinate system: gx, gy, and gz. For complexes with an axial symmetry elongated along the z-axis, which is the case for our CuIIP in frozen state, gx = gy > gz The two equivalent g values is called as g⊥ while the other value is known as g‖. This has been clarified in the text The recorded EPR spectra are classical for mononuclear square-planar-complexes with 4N coordination, with g// values higher than gê“• characteristic of an elongated Jahn-Teller distortion around the CuII center.
- Page 5, line 158 to 160: For the overall experiment and for the statement – ‘For all CuII(P) complexes, such so low cathodic potential values indicate that the Cu(P) complexes should resist to reduction by Asc (below as certain concentration)’. Additional controls could be more conclusive. For example- Choose the best ‘peptide -CuII complex’, treat with Asc and measure UV-vs, EPR and CV.
Author reply:
As the reviewer rightly suggested, we recorded EPR and UV-vis spectrum of CuIIWHHG with addition of 1eq of Ascorbate. In EPR the intensity of the spectrum can change depending on how the tube in placed in the cavity. However, the UV-vis experiment in clear-cut enough to say that CuII(P) is not reduced by Asc. Indeed, the intensity of the d-d band, reminiscent of the CuII(P), doesn’t change before and after addition of ascorbate.
Figure supp. Low-temperature (120 K) X-band EPR spectra of CuII(P) + Ascorbate : WHHG (blue curve), [65CuII] = 500 μM, [P] = 500 μM, [Ascorbate]= 500 µM, [HEPES] = 50 mM, pH 7.4; 10% (v/v) of glycerol as a cryoprotectant. T = 120 K, n ≈ 9.5 GHz, mod. ampl. = 5 G, microwave power: 5 mW.
SEE Attached file
Figure supp. Uv-vis spectra of WHHG-CuII (red curve) WHHG-CuII + 1eq of Ascorbate (yellow curve). Experimental conditions: [CuII] = [P]= [Asc] = 400 µM, [HEPES] = 100 mM, pH 7.4, T = 25 °C
SEE Attached file
- Page 6, 181: Please enlist CuII affinity value of ATCUN peptides as one table in SI.
Author reply:
The range of affinity of the ATCUN peptides for CuII in given in the introduction: “In such environment, the CuII center is tightly bound with affinity about 1012 to 1014 M−1”
- Page 7, Figure 5: given legends on figure are VERY confusing. For example- Dark black represents CuIIAβ however, labeling (dark black) = CuIIAβ, CuVGHW (what does this CuVGHW labelled along with the black line? Likewise, the same for all curves. Also, there is no explanation about the 3rd faint/dotted line (faint orange and faint blue).
Author reply:
Explanation of the faint line has been added. All the captions have been revised to be more understandable.
- Page 8, Figure 6: Indicate the ½ life (s) for inset of panel C (same as in panel B).
Author reply:
The ½ life (s) in panel C is to fast to be measured
- Page 9, Scheme 3: Drawn scheme and below legend information doesn’t match especially, info of panel C, panel D and panel E. Careful revision/labelling is needed.
Author reply:
Careful revision and labelling have been conducted.
- Page 10, Figure 7: Panel A labelling- revise ‘ATCUN’ to ‘ATCUN 30s incubation’; Panel B labelling- revise ‘ATCUN incubated’ to ‘ATCUN 30min incubation’
Author reply:
The figure has been modified accordingly.
- Page 11, line 262: language correction may need. Three main experiments can be performed. Does this means proposed experiments and experiments that are done in this study?
Author reply:
The sentence “Three main experiments can be performed” have been changed to “Three experiments were performed”
- Page 13, line 370: remove ‘thank to the’
Author reply:
It has been modified in the revised version
- Page 13, line 373: similarly, please revise the sentence 373.
Author reply:
The sentence has been revised.
- Page 13, line 390: Please provide the peptide purity by HPLC and include ESI-MS of all synthesized peptides.
Author reply:
The ESI-MS of all the peptide have been added in the SI.
- References: keep consistent format. For example- in some case range of page number is given while in other cases it’s not.
Author reply:
Changes have been made to keep a consistent format for references.

Round 2
Reviewer 2 Report
The manuscript was revised, taking into account the proposed suggestions!
I suggest keeping the information about "optimal conditions" determined/optimized for the Fluorescence Spectroscopy and Cyclic Voltammetry steps and, validation.
Reviewer 3 Report
Almost all the concerns are carefully and satisfactorily addressed. I thank authors for revising manuscript carefully. The revised manuscript is now publishable in this journal while authors may consider to include mentioned suggestion.
Minor suggestion (highlighted in green): The obtained figures could be good addition in SI.
